# Automatic Segmentation for Favourable Delineation of Ten Wrist Bones on Wrist Radiographs Using Convolutional Neural Network

**DOI:** 10.3390/jpm12050776

**Published:** 2022-05-11

**Authors:** Bo-kyeong Kang, Yelin Han, Jaehoon Oh, Jongwoo Lim, Jongbin Ryu, Myeong Seong Yoon, Juncheol Lee, Soorack Ryu

**Affiliations:** 1Department of Radiology, College of Medicine, Hanyang University, Seoul 04763, Korea; msbbogri@hanyang.ac.kr; 2Machine Learning Research Center for Medical Data, Hanyang University, Seoul 04764, Korea; yoon5690@naver.com (M.S.Y.); doldoly@hanyang.ac.kr (J.L.); 3Department of Computer Science, Hanyang University, 222 Wangsimni-ro, Seongdong-gu, Seoul 04763, Korea; gdf1845@gmail.com; 4Department of Emergency Medicine, College of Medicine, Hanyang University, 222 Wangsimni-ro, Seongdong-gu, Seoul 04763, Korea; 5Department of Software and Computer Engineering, Ajou University, Suwon 16499, Korea; jongbin.ryu@gmail.com; 6Department of Artificial Intelligence, Ajou University, Suwon 16499, Korea; 7Biostatistical Consulting and Research Lab, Medical Research Collaborating Center, Hanyang University, Seoul 04763, Korea; rsa4648@hanyang.ac.kr

**Keywords:** wrist, carpal bone, segmentation, deep learning, CNN

## Abstract

Purpose: This study aimed to develop and validate an automatic segmentation algorithm for the boundary delineation of ten wrist bones, consisting of eight carpal and two distal forearm bones, using a convolutional neural network (CNN). Methods: We performed a retrospective study using adult wrist radiographs. We labeled the ground truth masking of wrist bones, and propose that the Fine Mask R-CNN consisted of wrist regions of interest (ROI) using a Single-Shot Multibox Detector (SSD) and segmentation via Mask R-CNN, plus the extended mask head. The primary outcome was an improvement in the prediction of delineation via the network combined with ground truth masking, and this was compared between two networks through five-fold validations. Results: In total, 702 images were labeled for the segmentation of ten wrist bones. The overall performance (mean (SD] of Dice coefficient) of the auto-segmentation of the ten wrist bones improved from 0.93 (0.01) using Mask R-CNN to 0.95 (0.01) using Fine Mask R-CNN (*p* < 0.001). The values of each wrist bone were higher when using the Fine Mask R-CNN than when using the alternative (all *p* < 0.001). The value derived for the distal radius was the highest, and that for the trapezoid was the lowest in both networks. Conclusion: Our proposed Fine Mask R-CNN model achieved good performance in the automatic segmentation of ten overlapping wrist bones derived from adult wrist radiographs.

## 1. Introduction

Acute wrist pain related to trauma or non-trauma causes is a common complaint presented in primary care and emergency rooms [1,2]. Imaging is often necessary to make a definitive diagnosis of wrist pain, along with access to a clear history and physical examination, because no predetermined decision is possible. Standard plain radiographs are used as the initial diagnostic radiologic evaluation for most patients with wrist pain [3,4,5]. However, it is difficult for physicians—even an experienced radiologist—to accurately identify each bone contour, and to interpret subtle changes, because the wrist is composed of ten bones—eight carpal bones (trapezium, trapezoid, capitate, hamate, pisiform, triquetrum, lunate, and scaphoid) and two long bones (distal radius and distal ulna)—that overlap each other [4,5] (Figure 1a). Although wrist radiographs provide limited information, due to the limitations of projection views and clinical observations, they still offer crucial support for diagnostic and therapeutic determination in clinical practice [3,4,5,6].

Recently, the segmentation of bones using computer-aided algorithms has been studied for use in clinical diagnosis and treatment planning [7,8,9,10]. Wrist bone segmentation also has been studied as a predecessor to wrist fracture classification [11], bone age assessment [12,13], and the diagnosis of rheumatoid arthritis [14,15,16]. However, most wrist bone segmentation methods use conventional mathematical methods. Gou et al. conducted automatic segmentation through a dynamic programming algorithm [17], and Manos et al. employed the region growing [18] and region merging algorithms sequentially after pre-processing, using a Canny edge detector [19]. In addition, some advanced algorithms have been applied to overcome the disadvantages related to each medical image domain by combining these conventional methods [20,21].

Few studies of wrist bone segmentation via wrist radiograph, using deep learning, have been reported for reasons such as the low contrast between bone and tissue, the distances between the carpal bones, and the bones’ irregular shapes. Moreover, there were fewer than ten wrist bones being segmented in these studies, because they were focused on bone age assessment in young children, whose wrist bones have not yet matured [22,23].

This study aimed to develop and validate an automatic segmentation algorithm for the prediction of the boundaries of ten wrist bones overlapping each other on an adult wrist radiograph, using a convolutional neural network.

## 2. Methods

### 2.1. Study Design

This was a retrospective study using wrist posteroanterior (PA) or anteroposterior (AP) radiographs, which were performed at one tertiary hospital (Seoul, Korea) between April 2020 and September 2021. This study was approved by the Institutional Review Board of Hanyang University Hospital, and the requirement for informed consent was waived by the IRBs of our hospital. All methods and procedures were carried out in accordance with the Declaration of Helsinki.

### 2.2. Dataset of Participants

We sorted and gathered wrist PA or AP radiographs from adult patients with wrist pain who visited the emergency room at the Hanyang University Hospital between January 2011 and December 2019. Their radiology reports stated “unremarkable study”, “non-specific finding”, or “no definite acute lesion”. Radiographs were excluded when accurate delineation was impossible as a result of screws or other implants, the severe deformation of anatomical structures caused by acute fractures in another area, or past damage, malformation, or casts. Labeling for the ground truth masking of wrist bones was conducted with a program that was self-made and customized using a tool implemented in Matlab R2018b (MatLab, MathWorks, Natick, MA, USA), as shown in Figure 1b. The process was as follows: (1) classification as one of ten wrist bones; (2) delineation of each bone’s boundary by two emergency physicians for segmentation, and (3) review and revision by a radiologist. Radiograph images were extracted using the picture archiving and communication system (PACS, PiView, INFINITT Healthcare, Seoul, Korea) as digital imaging and communication and medicine (DICOM)-format images, and stored in the Joint Photographic Experts Group (JPEG) format. No personal information was included in the images used for data collection, with personally identifying data excluded. In addition, arbitrary numbers were assigned to the images, which were then coded and managed.

### 2.3. Data Pre-Processing

We pre-processed our dataset via three methods in order to train our network stably. First, the wrist directions in all the training images were corrected to leftwards, as all right-hand wrist images were horizontally mirrored. This strategy eliminated wrist direction variations and unnecessary computations. Secondly, the sizes of the wrist radiography images were fixed, since the image size was different for every person. Finally, the input images were normalized, which is essential in order to effectively fine-tune the pre-trained Mask R-CNN (an object detection and segmentation simultaneously based on deep CNN) [24].

### 2.4. Network Architecture

The overall workflow of our method (Fine Mask R-CNN) for the automatic segmentation of wrist bones is illustrated in Figure 2a. It consists of two main steps—wrist region of interest (ROI) detection and segmentation.

#### 2.4.1. Wrist ROI Detection Model Using Single-Shot Multibox Detector (SSD)

In this paper, a specific region whose X-ray image only includes our target wrist bones is called the wrist ROI. The first stage is to detect the wrist ROI, then the segmentation model uses this detected wrist ROI as the input image. This cascade system can focus on this ROI and segment ten wrist bones more precisely, which could be helpful for our study, wherein the ROI is a small section of the overall image [25,26].

Here, we trained the Single-Shot Multibox Detector (SSD) [27], which is a network that performs object detection on all the feature maps of different sizes through multiple convolutional layers. Using these variously sized feature maps, SSD can detect small to large objects effectively. Training this detection network requires the ground truth bounding box of ROI, which we manually labeled according to the rule [23]. The size of the ROI is the average of 620 × 470 pixels, which is about one-third the size of the original image. This extracted ROI was re-scaled to 1820 × 1450 pixels, and then used as an input for the wrist segmentation model.

#### 2.4.2. Wrist Segmentation Model Using Mask R-CNN with the Extended Mask Head

Most segmentation studies based on deep learning [7,28,29,30] have used the U-Net [31] architecture, which is the most popular algorithm for biomedical image segmentation. However, our proposed segmentation model was based on the Mask R-CNN [24], which is widely used for instance segmentation because most adult wrist bones overlap each other, especially the eight carpal bones. Therefore, some of the pixels could include two or more types of wrist bones.

In this paper, a finer segmentation network was proposed, which modified the mask head of the Mask R-CNN in two ways. Our first contribution to network design was a larger input size of the mask head. We used the 28 × 28 input feature, which is larger than the original, in the Mask R-CNN, as shown in Figure 2b. This advancement was motivated by the blurry contour problem mentioned in [32], which pointed out that the blurry contour appears as a result of a low-resolution regular grid of segmentation method. In other words, in the process of resizing the size of this coarse output to the original ROI size, the details nearby object boundaries were over-smoothing. We expect that a larger input size achieves better performance, but this will depend on the limitation of the hardware resource. Since we used the 28 × 28 input feature, the output probability map used for the segmentation model is 28 × 28 × 11, wherein 11 is the number of classes (ten wrist bones and background). We can prevent over-smoothing when interpolating the output probability map to the original image size by using this larger possibility map. This approach could be effective for use on the distal radius and distal ulna, because the resolution of these bones is greater than eight carpal bones. Additionally, we changed a mask head architecture from the original to an encoder–decoder structure, motivated by the U-Net, shown in Figure 2b. The structure of the U-Net connects multiple feature maps of different scales in an encoder–decoder architecture. With this architecture, high-resolution features can be combined with the upsampled features and more precise segmentation performance can be achieved than our baseline network, Mask R-CNN. 

The weights of our proposed mask branch are updated similarly to those of the Mask R-CNN. A mask probability map $\hat{y}$ is computed using a per-pixel sigmoid function at each output pixel value. Then, the binary cross-entropy loss of each mask probability map $\hat{y}$ for N ROIs is computed. The final mask loss $L_{mask}\left(\hat{y}\right)$ is computed as
(1)$$BCE\left(y,\;\hat{y}\right)=-\overset{2}{\underset{i=1}{\sum }}\;\left[y_{i}log\hat{y_{i}}+\left(1-y_{i}\right)loglog\;\left(1-\hat{y_{i}}\right)\;\right],\;\;\;\;$$
(2)$$L_{mask}\left(\hat{y}\right)=\frac{1}{N}\overset{N}{\underset{k=1}{\sum }}\;BCE\left(y_{ck},\;\hat{y_{ck}}\right),\;\;$$
where $y_{i}$ is the ground truth class of either the bone or the background, and $y_{ck}$ and $\hat{y_{ck}}$ are the ground truth and probability map corresponding to the predicted class of the *k*th ROI, respectively. Since the bone boundary is a very difficult region to correctly segment, the improvement of segmentation quality in the boundary region is a significant achievement, and offers much better results than visual observation.

#### 2.4.3. Training and Validation of Automatic Segmentation Using Fine Mask R-CNN and Mask R-CNN

To ensure the consistency of our model, a five-fold cross-validation was employed in our experiments. With randomly divided wrist X-ray images into five parts, we used four out of five of which are used for training and the other for testing. Depending on which part we choose as the test dataset, we can have five different train/test dataset combinations. Therefore, we can train the five models with five different train and test datasets and analyze their outputs to ensure our model’s robustness. In addition, this evaluation process can perform subject-based cross-validation by using all bones of one person used only for training or testing in one training phase.

Two networks were trained using a stochastic gradient descent (SGD) optimizer with a momentum equal to 0.9, and the initial learning rates were 0.001 and 0.0075, with weight decay factors of 0.0005 and 0.0001, respectively. The overall system employed the Pytorch library, and all the training and testing phases were performed on a GeForce GTX 1080 Ti GPU (NVIDIA, Santa Clara, CA, USA).

The baseline network and our proposed network were pre-trained by the ImageNet dataset and were fine-tuned with our collected wrist X-ray dataset. The fine-tuning algorithm transfers network parameters learned from a large common dataset to a specific task. Various studies have used a fine-tuning algorithm to analyze medical images, and its effectiveness has also been proven in the detection of other wrist fractures [33,34]. 

### 2.5. Primary Outcomes and Quantitative Evaluation

Our primary outcome was an improvement in the delineation predicted by networks in compliance with each wrist bone’s ground truth masking. For the quantitative evaluation of performance, we used the Dice coefficient (Dice), a well-known area-based metric for evaluating segmentation algorithms. It estimates the degree of overlap between the ground truth area and the predicted area. The Dice coefficient is calculated as follows:(3)$$\mathrm{Dice}=\frac{2\mathrm{TP}}{2\mathrm{TP}+\mathrm{FP}+\mathrm{FN}}$$

TP, FP and FN are the numbers of true positive, false positive, and false negative pixels, respectively. We measured the Dice coefficient for each bone, and calculated the average of 8 carpal bones, as well as the average Dice of 2 forearm bones and the total Dice of 10 bones to assess the overall performance of the model. This metric holds a value between 0 and 1, and higher values mean better predictions.

Additionally, we performed Turing tests on the ground truth masking performed by clinicians and the masking predicted by our network for the segmentation of ten wrist bones. The Turing test examines a machine’s ability to exhibit intelligent behavior indistinguishable from, or equivalent to, that of a human. One professor and two residents at the department of radiology, who were not authors, were blinded as to the subject vis a vis masking; they scored between 1 (worst) and 5 (best) in terms of the quality of the delineation of the segmentation boundaries on the ground truth mask and the predicted mask of 140 images.

### 2.6. Visualization of Predicted Masking of Wrist Bones through Automatic Segmentation by Networks

The Dice coefficient used for quantitative evaluation in this paper is frequently used for segmentation model evaluations; however, it is an area-based metric, so it has a disadvantage in that it cannot evaluate the accuracy of boundaries. Therefore, we visualized the wrist bone segmentation results using networks in order to yield explainable and insightful analyses.

### 2.7. Statistical Analysis

The data were compiled using a standard spreadsheet application (Excel 2016; Microsoft, Redmond, WA, USA) and analyzed using NCSS 12 (Statistical Software 2018, NCSS, LLC. Kaysville, UT, USA, ncss.com/software/ncss). Kolmogorov–Smirnov tests were performed to demonstrate the normal distribution of all the datasets. We generated descriptive statistics, and here present them as frequency and percentage values in the categorical data, and as either median and interquartile range (IQR) (non-normal distribution) or mean and standard deviation (SD) (normal distribution). Paired *t*-tests or Wilcoxon signed rank tests were used to compare the performance between the Mask R-CNN as the baseline network and the Fine Mask R-CNN as the proposed network, and the Turing test was used to compare between the ground truth and the predicted mask. *p*-values < 0.05 were considered statistically significant. The intraclass correlation coefficient (ICC) was used to determine the agreement between three evaluators used in the Turing test. Values of ICC less than 0.5, between 0.5 and 0.75, between 0.75 and 0.9, and greater than 0.90 were indicative of poor, moderate, good, and excellent reliability, respectively [35].

## 3. Results

In total, 702 images were collected from 702 patients and all images were labeled for the annotation and segmentation of ten wrist bones. The baseline characteristics of participants who provided labeled images were 45.74 (16.66) years old, 53.30% female, and 69.42% images of the left wrist. 702 labeled images split 140 of set A, 140 of set B, 140 of Set C, 141 of set D, and 141 of set E. All images used for training both wrist ROI detection and segmentation model were through five-fold validation, and we obtained the test results of 702 images for our proposed models. 

### 3.1. The Performance Test between the Fine Mask R-CNN and the Mask R-CNN for the Automatic Segmentation of Wrist Bones

The overall performance (mean [SD] of Dice) in the auto-segmentation of 10 wrist bones after training increased from 0.93 (0.01) via the Mask R-CNN to 0.95 (0.01) via the Fine Mask R-CNN (*p* < 0.001). All values for each bone were higher in the Fine Mask R-CNN than in the Mask R-CNN (all *p* < 0.001). The Dice value of the distal radius was the highest (0.96 (0.01)), and that of the trapezoid bone was the lowest (0.91 (0.05)) after training with the Fine Mask R-CNN, whereas the Dice value of the distal radius was the highest (0.94 (0.02)), and that of the trapezoid bone was the lowest value (0.90 (0.05)) after training with Mask R-CNN (Table 1).

### 3.2. The Turing Test between Ground Truth Masking by Clinicians and Masking Predicted by Fine Mask R-CNN for the Automatic Segmentation of Wrist Bones

The total scores (median [IQR]) of all ten wrist bones were 47 (38–50) via predicted masking and 48 (38–50) via predicted masking and 48 (41–50) via ground truth masking (*p* < 0.001). The evaluators estimated that the delineation of ground truth masking was better than that of predicted masking in each carpal bone (all *p* < 0.001), except for the trapezoid and scaphoid (*p* = 0.25, and *p* = 0.39 respectively). The scores of the distal radius and ulnar bones were also significantly different between the two masking methods (all *p* < 0.001). The ICC values amongst the evaluators were poor to moderate, in terms of both the ground truth and the predicted masking (Table 2).

### 3.3. Visualization of Predicted Masking for Wrist Bone Segmentation by Two Networks

The visualizations used for the delineation of eight carpal bones and two distal forearm segments, created by two different networks, are shown in Figure 3. Our proposed Fine Mask R-CNN achieves closer and more accurate delineation with ground truth masking than the other approach.

## 4. Discussion

In this study, we have proposed a Fine Mask R-CNN, and this model performed better, with a 0.95 (0.01) Dice coefficient for the segmentation of ten wrist bones, including eight carpal bones and two distal forearm bones, from wrist radiographs of people between 18 and 80 years old. Currently, there are two established neural network models specifically used for image segmentation in the computer imaging field: the fully convolutional neural network (FCN) and Mask R-CNN. Meng et al. reported that FCN could segment the carpal site with a Dice coefficient of 0.78 (0.06), using hand and wrist radiographs of people between 0 and 18 years old [36]. Su et al. reported that carpal bones were successfully detected with a high Dice coefficient of 0.976 using threshold processing and boundary detection on hand radiograph images. However, this was only tested on 30 representative images of non-overlapping carpal bones [37].

We have assessed the performance of two approaches to the segmentation of ten wrist bones. Faisal et al. found that the range of Dice coefficients for the segmentation of eight carpal bones was 0.83~0.94 when the locally weighted K-means variational level set was applied, whereas the range was 0.91~0.96 when Fine Mask R-CNN was employed in our study [22]. Goo et al. showed that the mean Dice coefficient of the automatic segmentation of the distal ulna and radius with dynamic programing was about 0.90, when using forearm radiographs [17], while we achieved a mean [SD] Dice of 0.96 (0.01) with Fine Mask R-CNN. The use of a fracture detection CNN without segmentation, based on a Dense-161, for distal radio-ulnar fractures on plain radiographs showed a sensitivity of 90.3%, with a specificity of 90.3% [38]. The sensitivity and specificity of the CNN without segmentation in terms of detecting distal radial fractures (using EfcientNet-B2 in frontal view and EfcientNet-B4 in lateral view wrist radiographs) were 98.7% and 100%, respectively [39]. The use of a segmentation and fracture detection CNN, based on a DenseNet-121, for the automated detection of scaphoid fractures on plain radiographs achieved a Dice coefficient of 0.974 (0.014) and a sensitivity of 78%, with a specificity of 84%. This network could achieve performance levels comparable to human observation in detecting scaphoid fractures on radiographs [11]. Our proposed network for the segmentation of ten wrist bones could assist in the automatic detection of various wrist bone fractures on wrist radiographs.

Most studies on wrist bone segmentation have used the wrist radiographs of young children. Wrist bones are formed during infancy, and increasingly overlap as their size increases [13,37,40]. In our study using Fine Mask R-CNN on adults’ wrist radiographs, the performance for scaphoid, capitate, hamate, and lunate bones achieved Dice coefficients of over 0.95, because these bones are relatively large, and the overlap area with other bones is relatively small. However, the Dice values of some were lower, such as 0.93 for the trapezium, 0.91 for the trapezoid, and 0.93 for the pisiform. This is because the trapezium and trapezoid overlap in almost all areas in men over 7 years of age and women over 5 years of age [13], and the trapezium, trapezoid, and pisiform wrist bones overlap on the wrist PA radiographs of adults [41]. We have proposed a two-stage method that extracts the ROI from a wrist X-ray image first, and then segments the 10 bones within to solve this problem. Additionally, in the segmentation module based on Mask R-CNN, using an encoder–decoder-type network, spatial information can be preserved. This helped us to improve the segmentation performance by using the preserved spatial information. However, the capacity for the delineation of overlapping bones, such as the trapezium, trapezoid, and pisiform, was still worse than the others. 

The Turing test is an important measure of how “intelligent” a deep learning model is. In a study on the automatic segmentation of a clinical target volume in rectal cancer patients, at least three out of ten clinicians thought that the predicted masking in this region was better than the ground truth masking [42]. This is the first study to carry out a Turing test on the automatic segmentation of ten wrist bones using wrist radiographs. The evaluators could not assign superiority between the masking predicted by our network and the ground truth masking performed by clinicians for the segmentation of two (trapezoid and scaphoid) wrist bones

Several limitations of this study should be considered. First, the data on the wrist radiographs and the patients originated from a single center, and our proposed model might not be suitable for other hospitals. Second, our proposed method was not an end-to-end network. Since Fine Mask R-CNN consists of two different neural networks—wrist ROI detection and wrist segmentation networks, the gradient cannot be shared directly between them. Therefore, our work needs to be extended to assess end-to-end networks that will establish a trainable attention module for future work. Finally, bias could not be eliminated from the Turing test because the test was performed by three radiologists from one center, without double blindness or randomization.

## 5. Conclusions

Our proposed CNN model exhibited a highly favorable performance in the automatic segmentation of ten overlapping wrist bones, consisting of eight carpal bones and the distal ulna and radius carpal bones, on plain wrist radiographs.

## Figures and Tables

**Figure 1 jpm-12-00776-f001:**
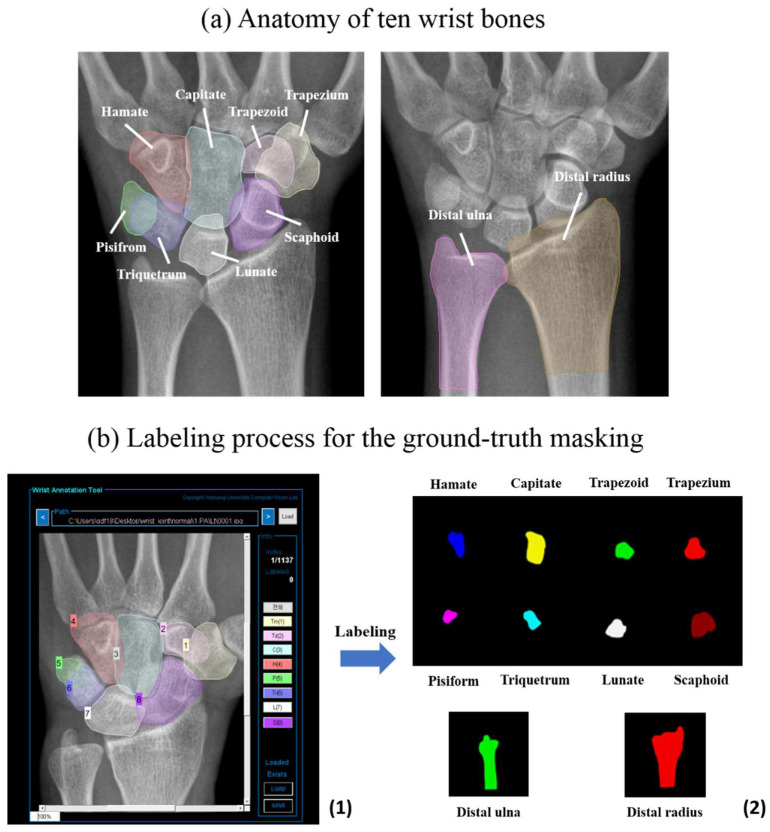
Anatomy and labeling of ten wrist bones on a wrist radiograph. (**a**) The anatomy of ten wrist bones, consisting of eight carpal bones and two distal forearm bones, on an anteroposterior radiograph. (**b**) Labeling process for the ground truth masking of wrist bones using a self-made customized tool. (**1**) The classification as one of ten wrist bones and the delineation of each bone’s boundary; (**2**) Labeling and extraction of each bone.

**Figure 2 jpm-12-00776-f002:**
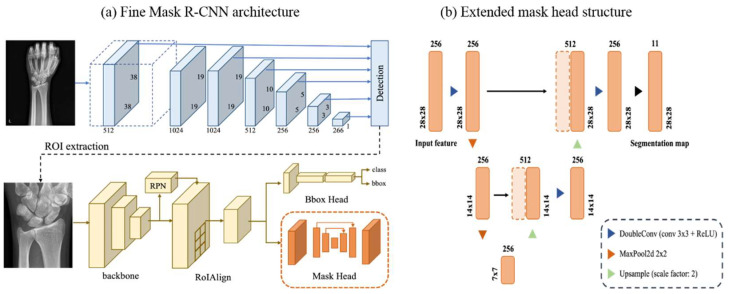
The Fine Mask R-CNN architecture. Our proposed network operates on a 2-stage method. (**a**) Detection of the regions of interest in the input wrist radiographs using SSD (blue) in the first stage and the delineation of 10 wrist bones using Mask R-CNN with the extended mask head in the second stage (yellow). (**b**) The structure of the extended mask head. This is an encoder–decoder structure, which can use previous information for the prediction of a specific part. ROI, region of interest; CNN, convolutional neural networks; SSD, Single-Shot Multibox Detector.

**Figure 3 jpm-12-00776-f003:**
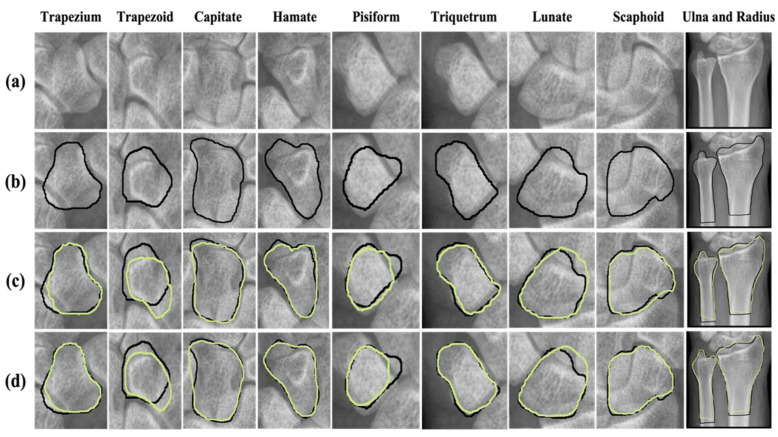
Visualization of Fine Mask R-CNN and Mask R-CNN network for the segmentation of ten wrist bones. (**a**) Original image of each wrist bone on the radiograph, (**b**) Delineation of segmented bone by physicians manually, (**c**) Delineation of segmented bone by Mask R-CNN, (**d**) Delineation of segmented bone by Fine Mask R-CNN with an extended mask head. Black lines indicate the ground truth masking segmented by physicians and yellow lines indicate the predicted masking segmented by CNN. CNN; convolutional neural networks.

**Table 1 jpm-12-00776-t001:** Comparison of the performance outcomes between the Mask R-CNN and the Fine Mask R-CNN for the automatic segmentation of ten wrist bones.

	Tm	Td	C	H	P	Tr	L	S	Carpal	R	U	Forearm	Total
Mask R-CNN Dice, mean [SD]	0.92 (0.03)	0.90 (0.05)	0.93 (0.04)	0.93 (0.02)	0.91 (0.05)	0.93 (0.02)	0.93 (0.02)	0.93 (0.02)	0.92 (0.01)	0.94 (0.02)	0.93 (0.02)	0.94 (0.01)	0.93 (0.01)
Fine Mask R-CNN Dice, mean [SD]	0.93 (0.03)	0.91 (0.05)	0.95 (0.04)	0.95 (0.02)	0.93 (0.04)	0.95 (0.02)	0.95 (0.02)	0.96 (0.02)	0.94 (0.01)	0.96 (0.01)	0.96 (0.02)	0.96 (0.01)	0.95 (0.01)
Comparison between two networks’ *p*-values	<0.001 *	<0.001 *	<0.001 *	<0.001 *	<0.001 *	<0.001 *	<0.001 *	<0.001 *	<0.001 *	<0.001 *	<0.001 *	<0.001 *	<0.001 *

Dice, Dice coefficient; SD, standard deviation; Tm, trapezium; Td, trapezoid; C, capitate; H, hamate; P, pisiform; Tr, triquetrum; L, lunate; S, scaphoid; R, distal radius; U, distal ulna. Paired *t*-tests were used to compare the performance between two networks according to normality. * *p*-values < 0.05 were considered statistically significant.

**Table 2 jpm-12-00776-t002:** Result of the Turing test between the ground truth masking segmented by clinicians and the predicted masking segmented by Fine Mask R-CNN for the automatic segmentation of ten wrist bones.

			Tm	Td	C	H	P	Tr	L	S	Carpal	R	U	Forearm	Total
Prediction	Score	Median	4	5	4	5	5	5	5	5	37	5	5	10	47
		IQR	4, 5	5, 5	4, 5	4, 5	4, 5	4, 5	5, 5	5, 5	36, 38	5, 5	5, 5	9, 10	45, 48
	ICC	Mean	0.58	0.59	0.60	0.60	0.54	0.77	0.71	0.31	0.51	0.61	0.51	0.56	0.54
		95% CI	0.45, 0.69	0.46, 0.70	0.47, 0.70	0.46, 0.70	0.39, 0.65	0.70, 0.83	0.62, 0.78	0.10, 0.48	0.35, 0.64	0.48, 0.71	0.36, 0.63	0.42, 0.67	0.36, 0.66
Ground Truth	Score	Median	5	5	5	5	5	5	5	5	39	5	5	10	48
		IQR	4, 5	5, 5	4, 5	5, 5	5, 5	5, 5	5, 5	5, 5	37, 39	5, 5	5, 5	10, 10	47, 49
	ICC	Mean	0.57	0.04	0.39	0.56	0.42	0.61	0.55	0.52	0.48	0.65	0.40	0.57	0.54
		95% CI	0.36, 0.70	0.25, 0.27	0.21, 0.54	0.42, 0.67	0.24, 0.56	0.49, 0.71	0.41, 0.66	0.36, 0.64	0.27, 0.63	0.54, 0.74	0.22, 0.55	0.44, 0.68	0.34, 0.67
Score between two maskings		*p*-value	<0.001 *	0.25	<0.001 *	<0.001 *	<0.001 *	<0.001 *	<0.001 *	0.39	<0.001 *	<0.001 *	<0.001 *	<0.001 *	<0.001 *

IQR, interquartile range; ICC, intraclass correlation coefficient; Tm, trapezium; Td, trapezoid; C, capitate; H, hamate; P, pisiform; Tr, triquetrum; L, lunate; S, scaphoid; R, distal radius; U, distal ulna. The Wilcoxon signed rank test was used to compare the Turing test results between the prediction and the ground truth masking. * *p*-values < 0.05 were considered statistically significant. Values of ICC less than 0.5, between 0.5 and 0.75, between 0.75 and 0.9, and greater than 0.90 were indicative of poor, moderate, good, and excellent reliability, respectively.

## Data Availability

The data presented in this study are available on request from the corresponding author.

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
