# Peer review of "Automatic Segmentation for Favourable Delineation of Ten Wrist Bones on Wrist Radiographs Using Convolutional Neural Network"

_jpm, 2022, doi:10.3390/jpm12050776_

Round 1

Reviewer 1 Report

The article entitled "Automatic Segmentation for Favourable Delineation of Ten Wrist Bones on Wrist Radiographs Using Convolutional Neural Network" is a well-written application paper that addresses the specific problem of boundary delineation of forearm bones. 

In my opinion, there is nothing special about their methodology; their convolutional neural network approach is adequate and quite standard,
in other words, I do not see anything unusual or especially new in the data treatment. Equally the obtained results are fully expected.

Author Response

Thank you for your review and point

Reviewer 2 Report

Summary:    ​This work collected wrist posteroanterior (PA) or anteroposterior (AP) radiographs to generate the dataset.It also proposed a method called the Fine Mask R-CNN which use Single-Shot Multibox Detector (SSD)  to achieve the wrist region of interest (ROI) detection, and modify the mask head  with the 28×28 input size and an encoder–decoder structure motivated by U-Net to achieve more effective prediction of the boundaries of ten wrist bones overlapping each other on an adult wrist radiograph.   Advantages:   1. This work is a pioneering solution to the problem of segmenting carpal borders with overlapping problems. 2. This work achieved more effective segmentation of wrist radiographs. 3. This work first carried out a Turing test on the automatic segmentation of ten wrist bones using wrist radiographs.  4. This work compared Masked R-CNN and Fine Masked R-CNN through five-fold validations to ensure the consistency of the model.   Disadvantages:   1. The wrist radiographs used for model training were originated from a single center, the generalization ability of the trained models are yet to be considered. 2. Detection and segmentation network were trained separately, gradient can not be shared directly. 3. Bias could not be eliminated from the Turing test because the test was performed by three radiologists from one center, without double blindness or randomization.  4. The enhancement of the Dice coefficient for quantitative assessment seems to be small, only about 0.01-0.02 for each type of bones, although the paper said that it cannot evaluate the accuracy of boundaries well.   Question:   1. Did you consider adding supplementary comparison experiments? For example, FCN, etc.? 2. Why did you not consider joint training in the beginning? How would the inability to share gradients affect your experiments besides the increase in computational effort? 3. Why did you choose 28*28 as the input feature size of the Mask Head? 4. As you described in 2.4.3, the baseline segmentation network was fine-tuned by the ImageNet dataset, so the wrist PA or AP radiographs dataset you collected was only used for wrist ROI detection network training? But in *2.3*, you said that normalized your input images can help fine-tune the pre-trained Mask R-CNN ? Or do you want to express "the model is pre-trained by ImageNet?"

Author Response

Thank you and reviewers for your kind and careful comments.

Reviewer 3 Report

1. It would be important to know the average dice similarity coefficient (DICE) that in your wrist segmentation model; Whether it is comparable to manually drawn label.

2. It would be important to know the exact numbers the the labeled images  that used for training, development, testing, and method validation stages.

3. Please mention how to performed subject-based cross-validation?

4. Please indicate the hardware and software using in this study.

5. Please detailed mention the evaluation process of the network training using the “hold-out” approach.

6 Please add the manual segmentation images in Figure 3 as examples.

Author Response

Thank you and reviewers for your kind and careful comments
